# Relationship between anion gap and 28-day all-cause mortality in patients with acute pulmonary edema: A retrospective analysis of the MIMIC-IV database

Ping Guo[1], Yuwen Liu[2], Xiaomi Huang[3], Yanfang Zeng[2], Zhonglan Cai[4], Guang Tu [4]*

1 Department of ICU, Xinfeng County People's Hospital, Ganzhou, China, 2 Department of Cardiovascular Medicine, Suizhou Hospital, Hubei Medicine University, Suizhou, China, 3 Department of Cardiology, Qixingguan District People's Hospital, Bijie, China, 4 Department of Cardiology, Lichuan People's Hospital, Fuzhou, China

* tuguang060666@163.com

## Abstract

### Background

Acute pulmonary edema is a severe clinical condition with high mortality. The anion gap, reflecting metabolic acid-base disturbances, is often elevated in critically ill patients. However, its relationship with outcomes in acute pulmonary edema remains unclear.

### Objective

To explore the association between admission anion gap levels and 28-day all-cause mortality in patients with acute pulmonary edema.

### Methods

This retrospective cohort study utilized data from the MIMIC-IV database (2008–2019) and included adult patients with acute pulmonary edema. Patients were categorized into quartiles based on anion gap levels. Cox regression models analyzed the relationship between anion gap and mortality, with restricted cubic spline (RCS) curves, Kaplan-Meier analysis, and subgroup analyses.

### Results

A total of 1094 patients were included. Univariate Cox regression showed a positive correlation between anion gap levels and 28-day mortality (HR = 1.13, 95%CI: 1.09–1.17, $P < 0.001$). Multivariate analysis confirmed anion gap as an independent predictor (HR = 1.11, 95%CI: 1.07–1.15, $P < 0.001$). The RCS curve indicated a non-linear relationship, and Kaplan-Meier analysis showed lower survival in higher anion gap groups ($P < 0.001$). Subgroup analysis revealed significant interactions between

**Data availability statement:** The data utilized in this study can be obtained from the Massachusetts Institute of Technology (MIT) and Beth Israel Deaconess Medical Center (BIDMC) by request. Access to these data is facilitated through the MIMIC-IV v3.1 database, which is publicly accessible at https://physio-net.org/content/mimiciv/3.1/. Researchers who wish to use these data should adhere to the guidelines detailed on the website to secure access.

**Funding:** The author(s) received no specific funding for this work.

**Competing interests:** The authors have declared that no competing interests exist.

**Abbreviations:** CI, confidence interval; HR, hazard ratio; MIMIC-IV, medical information mart for intensive care IV; RCS, restricted cubic spline; SBP, systolic blood pressure; DBP, diastolic blood pressure; WBC, white blood cell count; RBC, red blood cell count.

age and renal disease status, indicating that anion gap levels had a stronger association with mortality in younger patients and those without renal disease.

## Conclusion

Admission anion gap levels predict 28-day all-cause mortality in acute pulmonary edema patients, particularly in younger patients and those without renal disease. Clinically, anion gap monitoring should be emphasized, and individualized prognostic and treatment strategies should be developed with factors like age and renal status to improve outcomes.

## Introduction

Acute pulmonary edema is a critical clinical condition characterized by the rapid onset of respiratory distress due to fluid accumulation in the alveoli [1]. It is commonly associated with severe heart failure, particularly left ventricular dysfunction, and can lead to significant morbidity and mortality if not promptly managed [2]. The pathophysiology involves increased pulmonary capillary pressure, leading to the transudation of fluid into the lung interstitium and alveoli [3]. Early identification and intervention are crucial to improve outcomes, yet the clinical presentation can be variable and challenging to assess accurately. Notably, although many conditions can lead to an elevated anion gap, it may hold particular significance in patients with acute pulmonary edema. Prior studies have demonstrated an association between elevated anion gap levels and adverse outcomes in critically ill patients, and this association may serve as a marker of a significant systemic inflammatory response. Thus, exploring the relationship between anion gap levels and prognosis in patients with acute pulmonary edema is of great importance for better understanding the severity of the condition and guiding therapeutic decisions.

The anion gap is a widely used clinical parameter reflecting metabolic acid-base disturbances [4]. Elevated anion gap levels are often seen in critically ill patients and can indicate severe metabolic derangements [5]. However, the role of anion gap as a prognostic marker in patients with acute pulmonary edema remains less explored. Understanding this relationship could provide valuable insights into the severity of the condition and help guide therapeutic decisions.

The MIMIC-IV database offers a unique opportunity to investigate this association, given its comprehensive and detailed clinical data from a large cohort of patients [6,7]. By analyzing the data from this database, we aim to determine whether admission anion gap levels can predict 28-day all-cause mortality in patients with acute pulmonary edema.

## Methods

### Study design

This study is a retrospective cohort analysis utilizing data from the MIMIC-IV database. The primary objective was to investigate the association between admission anion gap levels and 28-day all-cause mortality in patients with acute pulmonary edema.

## Data source

The data were sourced from the MIMIC-IV database, which encompasses patient data from 2008 to 2019 [7]. The MIMIC-IV database is an anonymized dataset, and its use for research purposes is exempt from Institutional Review Board (IRB) approval due to its anonymized nature. However, to ensure ethical compliance, the study obtained approval from the PhysioNet IRB. Author Guang Tu completed the CITI Data or Specimens Only Research course, obtained approval for database access, and assumed responsibility for data extraction (certification number 65828445).

## Study population

The study population consisted of adult patients diagnosed with acute pulmonary edema within the MIMIC-IV database. Inclusion criteria included patients aged 18 years or older with a clinical diagnosis of acute pulmonary edema at admission and anion gap measurements within the first 24 hours of hospitalization. Patients with acute pulmonary edema were identified in the MIMIC-IV database using ICD codes J810 and 5184. The specific diagnostic criteria for acute pulmonary edema were not uniformly documented in the MIMIC-IV database, which reflects the variability in clinical practice. This limitation is inherent to the retrospective nature of the study. Exclusion criteria included patients with chronic pulmonary edema, cardiogenic shock, severe trauma, or major surgery prior to admission, as well as those with incomplete data (Fig 1).

## Data collection

Data were extracted from the MIMIC-IV database, including baseline demographic and clinical characteristics of the patients. These included gender, age, race, past medical history (such as myocardial infarction, heart failure, cerebrovascular disease, chronic lung disease, diabetes, renal disease, etc.), comorbidities, vital signs (systolic blood pressure, diastolic blood pressure, heart rate, etc.), laboratory test results (white blood cell count, red blood cell count, platelet count, hemoglobin levels, blood glucose, calcium, potassium, sodium levels, etc.), and anion gap levels. Patients with missing anion gap data were excluded. For other variables with missing values, if the missing proportion was greater than 50%, the corresponding patients were excluded. For other variables with missing values less than 50%, multiple imputation by chained equations was applied.

## Grouping method

Patients were divided into four groups (Q1-Q4) based on their anion gap levels [8,9]: Q1 group (anion gap < 10 mmol/dL), Q2 group (anion gap 10–12 mmol/dL), Q3 group (anion gap 12–15 mmol/dL), and Q4 group (anion gap > 15 mmol/dL).

## Statistical analysis

All statistical analyses were conducted utilizing R Statistical Software (Version 4.2.2, available at http://www.R-project. org, The R Foundation) and the Free Statistics Analysis Platform (Version 2.1, developed in Beijing, China, accessible via http://www.clinicalscientists.cn/freestatistics) [10]. Data were sourced from the MIMIC-IV database, which encompasses patient data from 2008 to 2019 [7]. The MIMIC-IV database is an anonymized dataset, and its use for research purposes is exempt from Institutional Review Board (IRB) approval due to its anonymized nature. The FreeStatistics package offers user-friendly interfaces for conducting standard analyses and visualizing data. R served as the core statistical processing component, while the graphical user interface was developed using Python. The majority of analyses could be executed with minimal clicks. This platform was crafted to facilitate reproducible research and interactive computational processes. A two-sided $p$-value threshold of less than 0.050 was set to determine statistical significance.

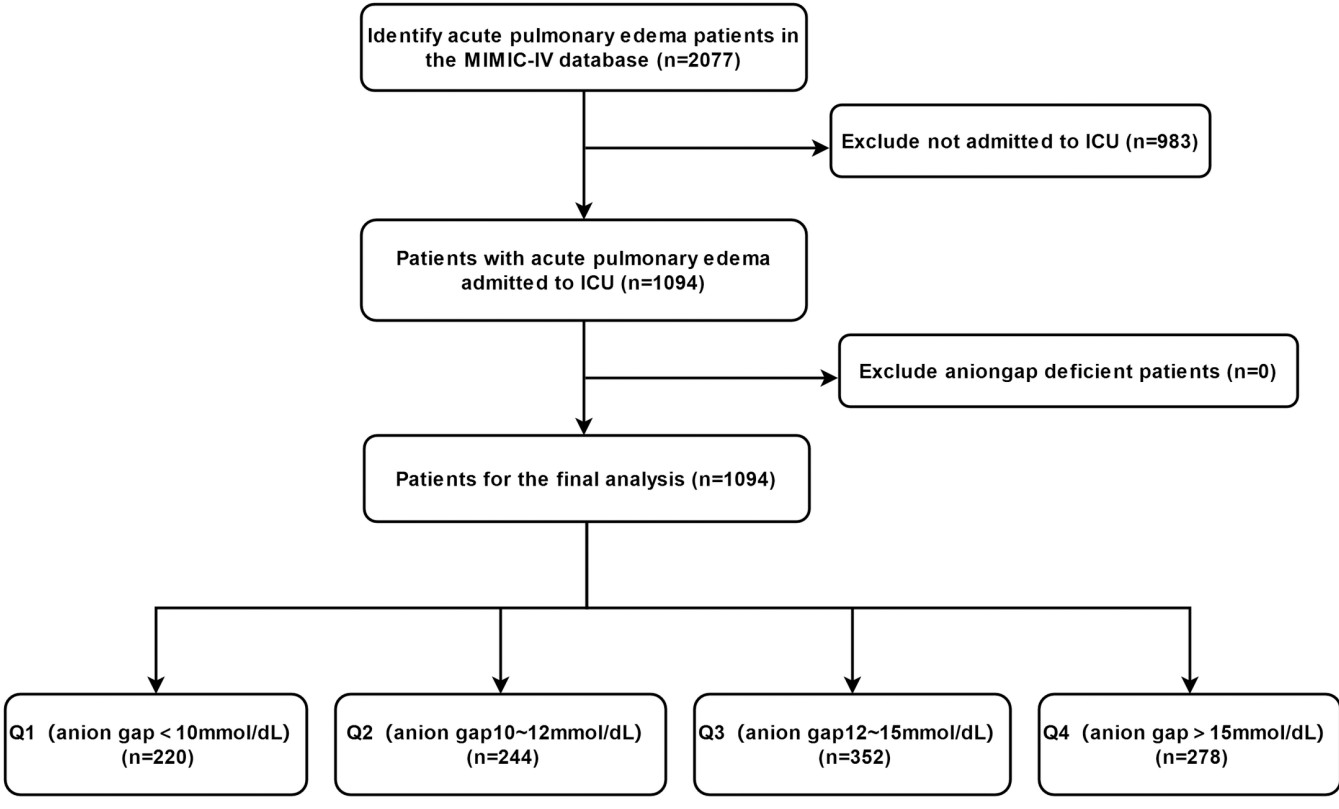

**Fig 1. Flowchart of patient inclusion.** The current chart illustrates the process of identifying 2077 acute pulmonary edema patients from the MIMIC-IV database, excluding 983 patients not admitted to the ICU, and including 1094 patients in the final analysis, with no patients excluded due to missing anion gap data.

No formal sample size calculation was performed for this retrospective analysis. The final sample size was determined based on the availability of data in the MIMIC-IV database that met the inclusion and exclusion criteria. Descriptive statistical analysis was performed on the patients' baseline characteristics. Univariate and multivariate Cox regression models were employed to analyze the relationship between anion gap levels and 28-day all-cause mortality. The multivariate Cox regression models were adjusted for age, gender, race, myocardial infarction, heart failure, cerebrovascular disease, chronic pulmonary disease, diabetes, renal disease, sapsii score, and sofa score. Restricted cubic spline (RCS) curves and Kaplan-Meier survival analysis were used to assess the impact of anion gap levels on mortality. In the restricted cubic spline analysis, the default settings of the statistical software were applied for the number and placement of knots. We conducted subgroup analyses to explore the stability of the relationship between anion gap levels and mortality across various clinical subgroups. These subgroups included age (<65 years vs. ≥65 years), gender (male vs. female), race (white, black, other), history of myocardial infarct (yes vs. no), history of heart failure (yes vs. no), cerebrovascular disease (yes vs. no), chronic pulmonary disease (yes vs. no), diabetes (yes vs. no), and renal disease (yes vs. no). Interaction terms were tested using Cox regression models to assess the significance of the interaction between anion gap levels and each subgroup variable on 28-day all-cause mortality. The interaction terms were calculated as the product of the anion gap variable and each subgroup indicator variable. A p-value less than 0.05 was considered statistically significant for the interaction terms.

## Results

### Baseline demographic and clinical characteristics

A total of 1094 patients with acute pulmonary edema were included. The mean age was 65.5 years, and there were no significant differences in gender distribution among the groups ($P=0.068$). The majority of patients were white (63.5%), and there were significant differences in systolic blood pressure ($P<0.001$) and heart rate ($P<0.001$) among the groups. Across quartiles, higher anion-gap groups exhibited incrementally greater renal injury (median creatinine $0.8 \rightarrow 1.4$ mg/dL; median BUN $14 \rightarrow 26$ mg/dL) and lower haemoglobin ($10.3 \rightarrow 9.6$ g/dL), while other laboratory parameters showed modest or non-systematic changes (Table 1).

### Anion gap as an independent risk factor for 28-day all-cause mortality

Univariate Cox regression analysis demonstrated a positive correlation between anion gap levels and 28-day all-cause mortality (HR = 1.13, 95%CI: 1.09–1.17, $P<0.001$). Multivariate Cox regression analysis confirmed anion gap as an independent risk factor for 28-day all-cause mortality (HR = 1.11, 95%CI: 1.07–1.15, $P<0.001$) (Tables 2 and 3).

### RCS curves and Kaplan-Meier curves

RCS analysis demonstrated a non-linear relationship between anion gap levels and 28-day all-cause mortality. Kaplan-Meier survival analysis further validated the impact of anion gap levels on survival prognosis, demonstrating that the survival rate of patients in the Q4 group was significantly lower than that of other groups ($P<0.001$) (Figs 2 and 3).

### Subgroup analysis and forest plot

Subgroup analysis revealed significant interactions between age and renal disease status. In patients aged <65 years, there was a stronger association between anion gap levels and mortality risk (HR: 1.25, 95% CI: 1.10–1.42, $P=0.002$), while this association was weaker in patients aged ≥65 years (HR: 1.05, 95% CI: 0.98–1.12). Similarly, in patients without renal disease, the association was significant (HR: 1.15, 95% CI: 1.08–1.22, $P<0.001$), while it was not significant in patients with renal disease (HR: 1.02, 95% CI: 0.95–1.09). Interaction tests confirmed significant interactions between age and anion gap levels ($P=0.018$) and between renal disease status and anion gap levels ($P=0.032$) (Fig 4).

## Discussion

Our study, through a retrospective cohort analysis, has revealed a significant association between admission anion gap levels and 28-day all-cause mortality in patients with acute pulmonary edema. Specifically, elevated anion gap levels were identified as an independent predictor of 28-day all-cause mortality, particularly in younger patients and those without renal disease. This finding not only underscores the potential clinical value of monitoring anion gap levels in patients with acute pulmonary edema but also suggests that clinicians should consider anion gap levels, along with age and renal function status, when assessing patient prognosis. By integrating these factors, clinicians can better identify high-risk patients and implement timely interventions to improve outcomes.

Although SOFA, SAPS II, lactate and BNP are established predictors, admission anion gap remained independently associated with 28-day mortality after adjustment for these variables, suggesting that it conveys additional prognostic information beyond that provided by conventional scores or lactate levels. Our study results are in line with several previous studies that have demonstrated a strong correlation between elevated anion gap levels and adverse outcomes in critically ill patients [9,11–13]. For instance, numerous studies have indicated that higher anion gap levels may reflect the severity of metabolic disturbances, especially in patients with acute heart failure and sepsis [14–16]. In our study, the association between anion gap levels and mortality remained significant even after adjusting for comorbidities and disease severity scores such as SOFA and SAPS II. This suggests that anion gap levels provide additional prognostic value

**Table 1. General characteristics of patients with acute pulmonary edema.**

| Variables | Total (n = 1094) | Q1 (n = 220) | Q2 (n = 244) | Q3 (n = 352) | Q4 (n = 278) | P _value |
|---|---|---|---|---|---|---|
| Gender, n (%) | | | | | | 0.068 |
| Female | 562 (51.4) | 97 (44.1) | 124 (50.8) | 186 (52.8) | 155 (55.8) | |
| Male | 532 (48.6) | 123 (55.9) | 120 (49.2) | 166 (47.2) | 123 (44.2) | |
| Race, n (%) | | | | | | 0.025 |
| White | 695 (63.5) | 145 (65.9) | 164 (67.2) | 227 (64.5) | 159 (57.2) | |
| Black | 97 (8.9) | 17 (7.7) | 16 (6.6) | 25 (7.1) | 39 (14) | |
| Orther | 302 (27.6) | 58 (26.4) | 64 (26.2) | 100 (28.4) | 80 (28.8) | |
| Age (year), mean (SD) | 65.5 ± 16.4 | 65.8 ± 15.5 | 65.9 ± 16.4 | 66.0 ± 16.1 | 64.4 ± 17.5 | 0.618 |
| SBP (mmHg), mean (SD) | 118.2 ± 17.0 | 115.7 ± 14.2 | 116.0 ± 14.6 | 119.0 ± 16.5 | 120.9 ± 20.6 | < 0.001 |
| DBP (mmHg), mean (SD) | 63.2 ± 11.3 | 61.6 ± 10.5 | 61.8 ± 10.5 | 64.1 ± 10.5 | 64.6 ± 13.2 | 0.003 |
| Heart rate(beats/min), mean (SD) | 87.7 ± 16.3 | 85.3 ± 14.6 | 85.2 ± 15.6 | 88.2 ± 16.3 | 91.1 ± 17.7 | < 0.001 |
| Myocardial infarct, n (%) | | | | | | 0.366 |
| No | 929 (84.9) | 180 (81.8) | 214 (87.7) | 298 (84.7) | 237 (85.3) | |
| Yes | 165 (15.1) | 40 (18.2) | 30 (12.3) | 54 (15.3) | 41 (14.7) | |
| Heart failure, n (%) | | | | | | 0.144 |
| No | 984 (89.9) | 200 (90.9) | 223 (91.4) | 321 (91.2) | 240 (86.3) | |
| Yes | 110 (10.1) | 20 (9.1) | 21 (8.6) | 31 (8.8) | 38 (13.7) | |
| Cerebrovascular, n (%) | | | | | | 0.014 |
| No | 944 (86.3) | 189 (85.9) | 196 (80.3) | 311 (88.4) | 248 (89.2) | |
| Yes | 150 (13.7) | 31 (14.1) | 48 (19.7) | 41 (11.6) | 30 (10.8) | |
| Chronic pulmonary, n (%) | | | | | | 0.542 |
| No | 817 (74.7) | 167 (75.9) | 179 (73.4) | 256 (72.7) | 215 (77.3) | |
| Yes | 277 (25.3) | 53 (24.1) | 65 (26.6) | 96 (27.3) | 63 (22.7) | |
| Diabetes, n (%) | | | | | | 0.256 |
| No | 844 (77.1) | 170 (77.3) | 198 (81.1) | 271 (77) | 205 (73.7) | |
| Yes | 250 (22.9) | 50 (22.7) | 46 (18.9) | 81 (23) | 73 (26.3) | |
| Renal disease, n (%) | | | | | | < 0.001 |
| No | 876 (80.1) | 190 (86.4) | 213 (87.3) | 285 (81) | 188 (67.6) | |
| Yes | 218 (19.9) | 30 (13.6) | 31 (12.7) | 67 (19) | 90 (32.4) | |
| WBC(×10⁹/L), Median (IQR) | 9.8 (6.8, 13.4) | 9.3 (6.9, 12.7) | 9.9 (6.9, 13.1) | 10.1 (6.8, 13.4) | 10.2 (6.4, 15.1) | 0.235 |
| RBC (×10⁹/L), mean (SD) | 3.5 ± 0.8 | 3.4 ± 0.8 | 3.5 ± 0.8 | 3.6 ± 0.8 | 3.4 ± 0.9 | 0.008 |
| Platelets(×10⁹/L), mean (SD) | 175.5 ± 101.7 | 165.5 ± 103.4 | 163.5 ± 80.5 | 187.5 ± 103.2 | 178.6 ± 112.9 | 0.013 |
| Hemoglobin(mg/dl), mean (SD) | 9.8 ± 2.3 | 9.3 ± 2.1 | 9.7 ± 2.2 | 10.3 ± 2.3 | 9.6 ± 2.5 | < 0.001 |
| Bun(mg/dl), Median (IQR) | 18.0 (12.0, 28.0) | 14.0 (10.0, 21.0) | 16.0 (12.0, 22.0) | 19.0 (12.8, 29.0) | 26.0 (15.2, 48.8) | < 0.001 |
| Creatinine(mg/dl), Median (IQR) | 0.9 (0.7, 1.4) | 0.8 (0.6, 1.0) | 0.9 (0.7, 1.1) | 1.0 (0.7, 1.4) | 1.4 (0.8, 2.8) | < 0.001 |
| Glucose(mmol/dl), mean (SD) | 120.6 ± 40.7 | 112.8 ± 27.5 | 122.1 ± 35.4 | 122.3 ± 35.8 | 123.1 ± 56.1 | 0.019 |
| Calcium(mmol/dl), mean (SD) | 8.1 ± 0.9 | 7.9 ± 1.0 | 8.0 ± 0.7 | 8.2 ± 0.8 | 8.1 ± 1.0 | < 0.001 |
| Potassium(mmol/dl), mean (SD) | 3.9 ± 0.6 | 3.9 ± 0.6 | 3.9 ± 0.5 | 3.9 ± 0.5 | 3.9 ± 0.7 | 0.622 |
| Sodium(mmol/dl), mean (SD) | 136.4 ± 4.9 | 135.9 ± 5.0 | 136.4 ± 4.4 | 136.7 ± 4.7 | 136.4 ± 5.3 | 0.292 |
| SAPSII(score), mean (SD) | 38.4 ± 13.5 | 38.9 ± 13.9 | 36.3 ± 12.1 | 37.1 ± 13.7 | 41.5 ± 13.6 | < 0.001 |
| SOFA(score), mean (SD) | 3.9 ± 2.1 | 3.8 ± 2.1 | 3.6 ± 1.9 | 3.8 ± 2.0 | 4.3 ± 2.4 | 0.003 |

**Table 2. A univariate Cox regression model evaluated the association between anion gap and 28-day all-cause mortality in patients with acute pulmonary edema.**

| Item | HR(95%CI) | P _value |
|---|---|---|
| Gender: male vs female | 0.97 (0.73,1.28) | 0.807 |
| Race: ref.=white | | |
| Black | 0.53 (0.27,1.05) | 0.07 |
| Orther | 1.38 (1.02,1.86) | 0.038 |
| Age (year) | 1.02 (1.01,1.03) | < 0.001 |
| SBP(mmHg) | 0.97 (0.96,0.98) | < 0.001 |
| DBP(mmHg) | 0.97 (0.96,0.99) | < 0.001 |
| Heart rate (beats/min) | 1.01 (1,1.02) | 0.018 |
| Myocardial infarct: yes vs no | 1.42 (1,2.04) | 0.053 |
| Heart failure: yes vs no | 1.15 (0.73,1.8) | 0.553 |
| Cerebrovascular: yes vs no | 1.07 (0.71,1.59) | 0.751 |
| Chronic pulmonary: yes vs no | 0.75 (0.53,1.06) | 0.101 |
| Diabetes: yes vs no | 1.03 (0.73,1.44) | 0.871 |
| Renal disease: yes vs no | 1.24 (0.89,1.73) | 0.21 |
| WBC(×$10^9$/L) | 1.01 (1,1.02) | 0.006 |
| RBC(×$10^{12}$/L) | 0.83 (0.7,0.99) | 0.04 |
| Platelets (×$10^9$/L) | 0.9987 (0.9972,1.0002) | 0.085 |
| Hemoglobin (mg/dl) | 0.93 (0.87,0.99) | 0.022 |
| Anion gap (mmol/dl) | 1.13 (1.09,1.17) | < 0.001 |
| Bun (mg/dl) | 1.02 (1.01,1.02) | < 0.001 |
| Creatinine (mg/dl) | 1.05 (0.98,1.13) | 0.194 |
| Glucose(mmol/dl) | 0.9999 (0.9964,1.0035) | 0.972 |
| Calcium(mmol/dl) | 0.91 (0.77,1.09) | 0.312 |
| Potassium (mmol/dl) | 0.9 (0.7,1.15) | 0.395 |
| Sodium (mmol/dl) | 1.03 (1,1.07) | 0.037 |
| SAPSII (score) | 1.05 (1.04,1.06) | < 0.001 |
| SOFA(score) | 1.18 (1.11,1.24) | < 0.001 |

**Table 3. A multivariate Cox regression model evaluated the association between anion gap and 28-day all-cause mortality in patients with acute pulmonary edema.**

| Variable | No | Crude model | | Model 1 | | Model 2 | | Model 3 | |
|---|---|---|---|---|---|---|---|---|---|
| | | HR (95%CI) | P _value | HR (95%CI) | P _value | HR (95%CI) | P _value | HR (95%CI) | P _value |
| Anion gap | 1094 | 1.13 (1.09~1.17) | <0.001 | 1.14 (1.1~1.18) | <0.001 | 1.14 (1.1~1.18) | <0.001 | 1.11 (1.07~1.15) | <0.001 |
| Anion gap (quartile) | | | | | | | | | |
| Q1 (<10) | 220 | 1(Ref) | | 1(Ref) | | 1(Ref) | | 1(Ref) | |
| Q2 (10~12) | 244 | 0.74 (0.41~1.32) | 0.305 | 0.73 (0.41~1.31) | 0.29 | 0.74 (0.41~1.32) | 0.302 | 0.84 (0.47~1.5) | 0.549 |
| Q3 (12~15) | 352 | 1.75 (1.11~2.77) | 0.016 | 1.74 (1.1~2.76) | 0.018 | 1.77 (1.12~2.81) | 0.015 | 1.98 (1.25~3.14) | 0.004 |
| Q4 (>15) | 278 | 2.74 (1.75~4.3) | <0.001 | 2.84 (1.81~4.47) | <0.001 | 2.89 (1.83~4.57) | <0.001 | 2.73 (1.72~4.32) | <0.001 |
| Trend.test | 1094 | 1.54 (1.33~1.78) | <0.001 | 1.56 (1.35~1.81) | <0.001 | 1.57 (1.35~1.82) | <0.001 | 1.5 (1.29~1.73) | <0.001 |

Footnote: Model 1 is the unadjusted model. Model 2 is adjusted for age, gender, and race. Model 3 is Model 2 plus adjustments for myocardial infarct, heart failure, cerebrovascular disease, chronic pulmonary disease, diabetes, renal disease, sapsii, and sofa.

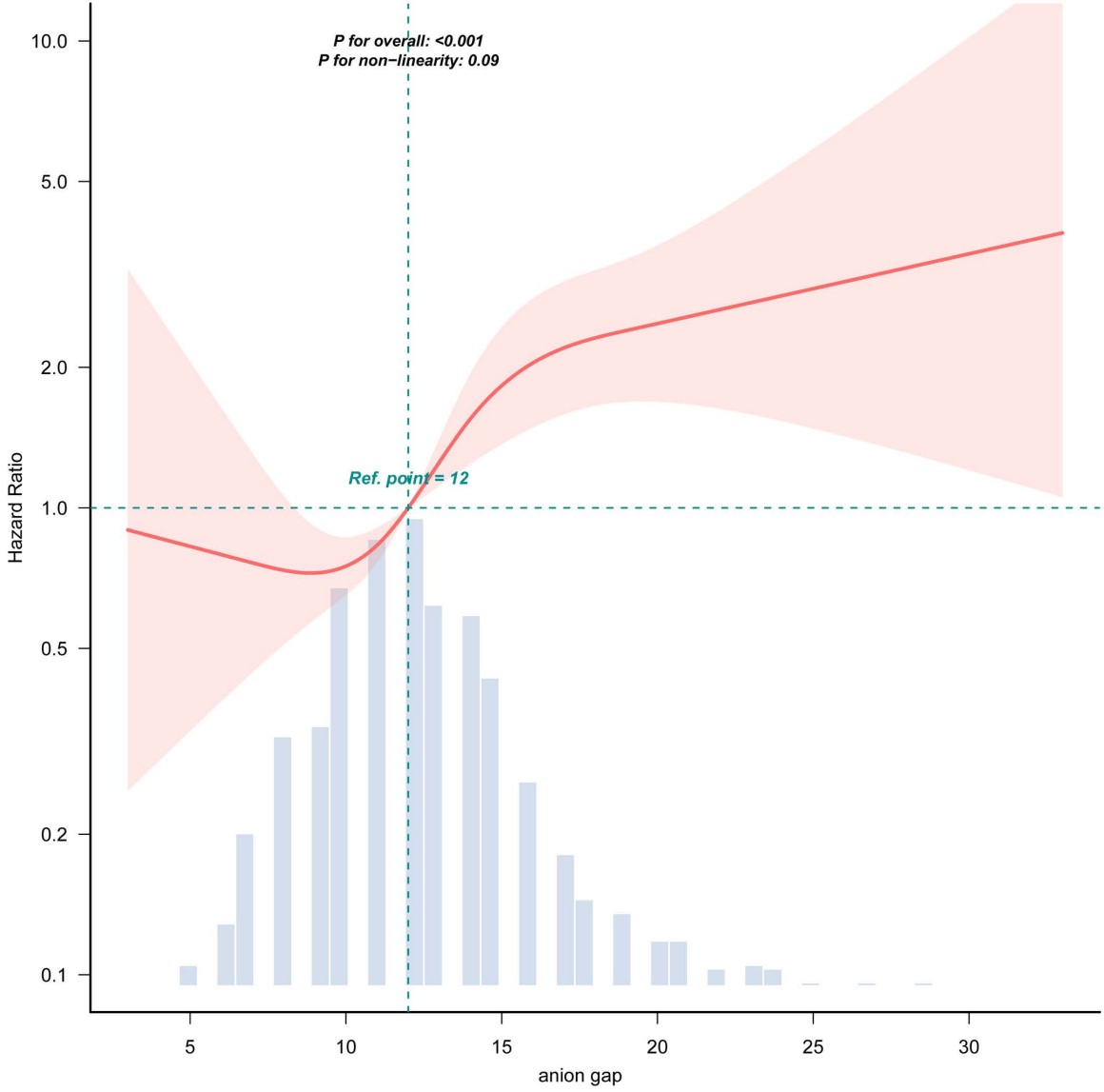

**Fig 2. RCS curve for the anion gap.** The current graph depicts the non-linear relationship between anion gap levels and 28-day all-cause mortality, with a reference point at 12 mmol/dL, showing an overall P-value of less than 0.001 and a non-linearity P-value of 0.09. anion gap (quartile): Q1(<10), Q2 (10~12), Q3 (12~15), Q4 (>15).

beyond traditional severity scores. These studies support the notion that the anion gap can serve as an indicator of metabolic disorders and potential organ dysfunction, further corroborating the association between anion gap levels and mortality in critically ill patients. However, our study specifically focuses on patients with acute pulmonary edema, adding new evidence to the literature by demonstrating that anion gap levels are independently associated with 28-day mortality in this specific patient population. This finding highlights the importance of anion gap monitoring in the context of acute pulmonary edema, which has not been extensively explored in previous studies. Additionally, some studies have suggested that elevated anion gap levels may be linked to systemic inflammatory responses and multi-organ dysfunction [17–19], which is consistent with the relationship between anion gap levels and mortality observed in our research. Collectively, these

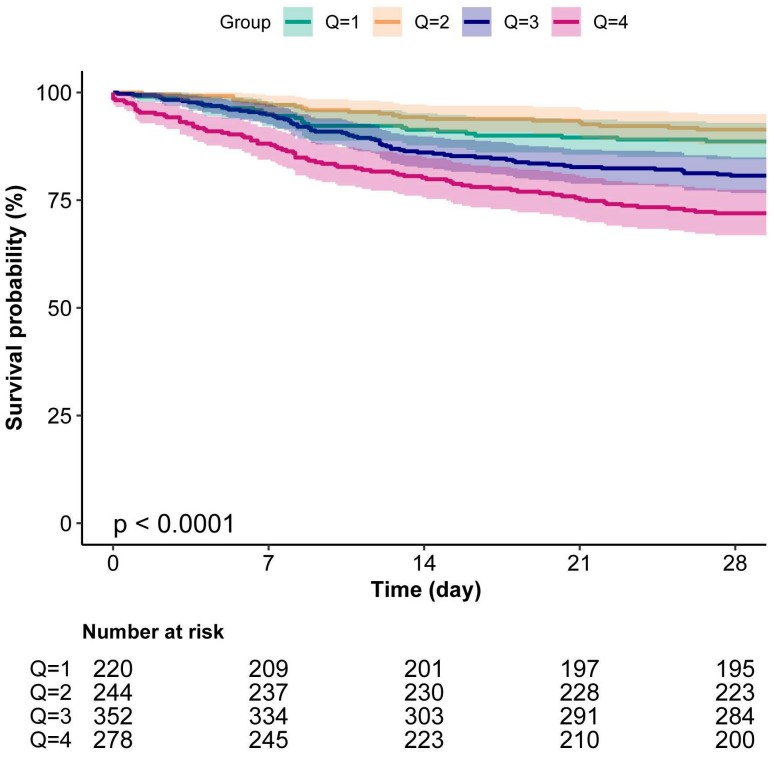

**Fig 3. Kaplan–Meier survival analysis curves for 28-day all-cause mortality.** The current chart presents the Kaplan-Meier survival analysis for 28-day all-cause mortality across quartiles of anion gap, with a P-value of less than 0.0001, and includes a table of numbers at risk at each time point below the graph.

studies concur that elevated anion gap levels are not only a marker of metabolic disturbances but also a potential indicator of the intensity of systemic inflammatory responses and the severity of organ dysfunction, thereby significantly influencing patient outcomes.

Elevated anion gap levels may influence the prognosis of patients with acute pulmonary edema through several mechanisms. First, an elevated anion gap typically indicates metabolic acidosis, which may result from increased anaerobic metabolism due to tissue hypoxia [20]. In patients with acute pulmonary edema, pulmonary congestion and impaired gas exchange can lead to inadequate tissue oxygenation, thereby triggering increased lactate production and a subsequent rise in the anion gap [21]. Second, inflammatory responses may also contribute to elevated anion gap levels [22]. The release of inflammatory mediators can lead to the accumulation of metabolic by-products, such as ketones, which further exacerbate metabolic acidosis [23]. Additionally, renal function plays a crucial role in maintaining acid-base balance, and impaired renal function can reduce the clearance of acidic metabolic by-products, leading to elevated anion gap levels [24,25]. Lastly, age may impact the relationship between anion gap levels and mortality. Younger patients generally have stronger physiological reserves, enabling them to better tolerate and recover from acute injuries, while older patients with multiple chronic comorbidities may obscure the impact of elevated anion gap levels [26–28]. These mechanisms interact with each other, collectively influencing the prognosis of patients with acute pulmonary edema. Therefore, in clinical practice, it is essential to consider these factors comprehensively to more accurately assess patient conditions and outcomes. The anion gap, as an easily measurable parameter, could potentially be incorporated into early risk stratification protocols or used to refine existing ICU scoring systems such as APACHE II or SOFA. The incremental value of anion gap in these scoring systems may lie in its ability to reflect metabolic disturbances and systemic inflammatory responses more

| Subgroup | Variable | Total | Event (%) | HR (95%CI) | | P for interaction |
|----------|----------|-------|-----------|------------|---|-------------------|
| **age** | | | | | | |
| <65 | aniongap | 495 | 72 (14.5) | 1.2 (1.14~1.26) | | 0.002 |
| >=65 | aniongap | 599 | 120 (20) | 1.08 (1.02~1.13) | | |
| **gender** | | | | | | |
| no | aniongap | 562 | 100 (17.8) | 1.12 (1.07~1.17) | | 0.595 |
| yes | aniongap | 532 | 92 (17.3) | 1.14 (1.09~1.2) | | |
| **race** | | | | | | |
| white | aniongap | 695 | 116 (16.7) | 1.11 (1.06~1.17) | | 0.514 |
| black | aniongap | 97 | 9 (9.3) | 1.17 (1.03~1.34) | | |
| orther | aniongap | 302 | 67 (22.2) | 1.15 (1.09~1.22) | | |
| **myocardial_infarct** | | | | | | |
| no | aniongap | 929 | 155 (16.7) | 1.12 (1.08~1.17) | | 0.379 |
| yes | aniongap | 165 | 37 (22.4) | 1.17 (1.07~1.27) | | |
| **heart_failure** | | | | | | |
| no | aniongap | 984 | 171 (17.4) | 1.14 (1.1~1.18) | | 0.173 |
| yes | aniongap | 110 | 21 (19.1) | 1.06 (0.96~1.17) | | |
| **cerebrovascular** | | | | | | |
| no | aniongap | 944 | 164 (17.4) | 1.13 (1.09~1.18) | | 0.537 |
| yes | aniongap | 150 | 28 (18.7) | 1.1 (0.99~1.21) | | |
| **chronic_pulmonary** | | | | | | |
| no | aniongap | 817 | 152 (18.6) | 1.14 (1.09~1.18) | | 0.416 |
| yes | aniongap | 277 | 40 (14.4) | 1.1 (1.01~1.19) | | |
| **diabetes** | | | | | | |
| no | aniongap | 844 | 148 (17.5) | 1.13 (1.08~1.17) | | 0.682 |
| yes | aniongap | 250 | 44 (17.6) | 1.14 (1.06~1.23) | | |
| **renal_disease** | | | | | | |
| no | aniongap | 876 | 147 (16.8) | 1.17 (1.12~1.21) | | <0.001 |
| yes | aniongap | 218 | 45 (20.6) | 1 (0.92~1.08) | | |

0.71    1.0    1.41
Effect (95%CI)

**Fig 4. Forest plot for the subgroup analysis of the relationship between hospital mortality and anion gap.** The current plot shows the relationship between anion gap and 28-day mortality across different subgroups, including age, gender, race, myocardial infarct, heart failure, cerebrovascular disease, chronic pulmonary disease, diabetes, and renal disease, along with their interaction P-values.

sensitively. By integrating anion gap levels into these scores, clinicians may be able to identify high-risk patients more accurately and implement timely interventions to improve outcomes.

In clinical practice, the anion gap can be pragmatically used as a readily available marker to assess the severity of metabolic disturbances in patients with acute pulmonary edema. Given its association with mortality, monitoring anion gap levels at admission and during hospitalization can help clinicians identify patients at higher risk for adverse outcomes. This information can be particularly valuable in resource-limited settings where more complex scoring systems may not be readily available. By incorporating anion gap into routine clinical assessments, clinicians can make more informed decisions regarding patient management and resource allocation.

## Limitations

Our study has several limitations. First, as a retrospective study, it is prone to selection bias and the influence of confounding factors. Although we adjusted for multiple potential confounders through multivariate analysis, there may still be unidentified or unadjusted confounding factors. Second, one significant limitation is the reliance on ICD codes for the diagnosis of acute pulmonary edema, rather than a standardized clinical or radiological definition. This approach may lead to misclassification bias, as the coding is dependent on the clinical judgment and documentation practices of healthcare providers. Additionally, the use of billing codes could potentially affect the accuracy of our findings, as these codes may not always reflect the true clinical picture. Future studies should consider incorporating standardized diagnostic criteria or additional validation methods to enhance the accuracy of the diagnosis. Furthermore, the specific diagnostic criteria for acute pulmonary edema were not uniformly documented in the MIMIC-IV database, which may introduce potential variability in the diagnosis of acute pulmonary edema among patients. This limitation is inherent to the retrospective nature of the study and the reliance on existing medical records. Third, our study relied solely on data from the MIMIC-IV database, which may have issues with data completeness and accuracy. Additionally, we were unable to obtain detailed clinical interventions for the patients, which could affect the assessment of prognosis. Third, there is a potential for misclassification bias in coding diagnoses, as the data are derived from an Electronic Health Record (EHR)-based database. This could lead to inaccuracies in the classification of acute pulmonary edema and other comorbidities. Lastly, our study did not delve into the specific mechanisms underlying elevated anion gap levels, and future research is needed to validate these mechanisms and explore targeted treatment strategies based on anion gap monitoring. Additionally, investigating the potential benefits of dynamic monitoring of anion gap levels over time, rather than relying on a single admission value, could provide more comprehensive prognostic information. Dynamic monitoring may help clinicians track changes in metabolic status and adjust therapeutic strategies accordingly, potentially improving patient outcomes. Moreover, since our study was limited to patients in the MIMIC-IV database, the results may not be fully generalizable to patient populations in other regions or different healthcare settings. Future prospective studies or multicenter validation studies are proposed to confirm these findings and assess the generalizability of our results to broader patient populations. Therefore, future studies should be conducted in broader patient populations to validate our findings and further explore their clinical application value.

## Conclusion

Our study indicates that admission anion gap levels predict 28-day all-cause mortality in patients with acute pulmonary edema, particularly in younger patients and those without renal disease. However, it is important to acknowledge that residual confounding from unmeasured factors such as lactate or albumin, which directly affect the anion gap, may still exist. Clinically, the anion gap could be incorporated into early risk stratification tools or ICU scoring systems to help identify high-risk patients more promptly. Future prospective studies or multicenter validation studies are proposed to confirm these findings and assess the generalizability of our results to broader patient populations. Further research is warranted to elucidate the underlying mechanisms and to develop targeted therapeutic strategies based on anion gap monitoring.

## Supporting information

**S1 Data.**

(XLS)

## Author contributions

**Conceptualization:** Guang Tu.

**Data curation:** Ping Guo, Yuwen Liu, Xiaomi Huang, Yanfang Zeng, Zhonglan Cai.

**Formal analysis:** Xiaomi Huang, Yanfang Zeng.

**Investigation:** Ping Guo, Yuwen Liu.

**Supervision:** Guang Tu.

**Writing – original draft:** Guang Tu, Ping Guo.

**Writing – review & editing:** Guang Tu.

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
