## [Decision Letter · Decision Letter 0]

19 Jun 2025

Dear Dr. tu,

Thank you for submitting your manuscript to PLOS ONE. After careful consideration, we feel that it has merit but does not fully meet PLOS ONE’s publication criteria as it currently stands. Therefore, we invite you to submit a revised version of the manuscript that addresses the points raised during the review process.

We look forward to receiving your revised manuscript.

Kind regards,

Eyüp Serhat Çalık

Academic Editor

PLOS ONE

Journal Requirements:

2. We note that your Data Availability Statement is currently as follows: 

“All relevant data are within the manuscript and its Supporting Information files”

**Additional Editor Comments:**

I congratulate the authors on this important work. The prognostic value of anion gap in patients with pulmonary edema has been well analyzed using the MIMIC-4 database. The manuscript has been reviewed by two reviewers, and their comments are provided below. Please address their comments appropriately and make the necessary revisions to your manuscript. We look forward to receiving your revised manuscript. Best success.

Reviewers' comments:

Reviewer's Responses to Questions

**Comments to the Author**

1. Is the manuscript technically sound, and do the data support the conclusions?

Reviewer #1: Yes

Reviewer #2: Yes

2. Has the statistical analysis been performed appropriately and rigorously?

Reviewer #1: Yes

Reviewer #2: I Don't Know

3. Have the authors made all data underlying the findings in their manuscript fully available?

Reviewer #1: Yes

Reviewer #2: Yes

4. Is the manuscript presented in an intelligible fashion and written in standard English?

Reviewer #1: Yes

Reviewer #2: Yes

Reviewer #1: This is a well-conducted and technically sound retrospective cohort study exploring the prognostic role of anion gap levels in patients with acute pulmonary edema using the MIMIC-IV database. The methodology is appropriate and clearly described, the statistical analyses are rigorous (including multivariable Cox regression, restricted cubic spline, Kaplan-Meier survival analysis, and subgroup interaction testing), and the conclusions are well-supported by the presented data.

Strengths of the study include:

Use of a large, publicly available and high-quality ICU dataset (MIMIC-IV).

Clear stratification of patients into quartiles of anion gap and appropriate adjustments for confounders.

Subgroup analysis exploring interactions with age and renal disease, which adds valuable clinical insight.

Suggestions for Improvement:

Language and Grammar: While the manuscript is overall intelligible and written in acceptable English, several minor grammatical and typographical issues persist. A language polishing service or editorial revision may enhance clarity and flow.

Figures and Tables: Ensure that all figures (e.g., Kaplan-Meier curves, RCS curves, forest plots) are of high resolution and clearly labeled for publication quality.

Ethics and Data Availability: While the MIMIC-IV database is exempt from IRB due to its anonymized nature, it may still benefit clarity to explicitly cite the IRB approval from PhysioNet and mention the CITI certification more formally within the Methods.

Limitations: The limitations are well addressed. However, a brief mention of potential misclassification bias in coding diagnoses (from EHR-based databases) would further strengthen the discussion.

Clinical Implications: Consider slightly expanding the clinical implications in the conclusion—for example, how the anion gap could be integrated into early risk stratification or ICU scoring systems.

Overall, this study contributes meaningfully to critical care and cardiopulmonary literature by identifying a simple, routinely measured biochemical marker (anion gap) as a significant predictor of short-term mortality in acute pulmonary edema. After minor editorial and stylistic revisions, it should be suitable for publication.

Reviewer #2: I have read this manuscript and I haver some questions. In the introduction, the authors do not tell me why this topic is important. Many conditions cause an anion gap. why should we be concerned about an elevated anion gap in a patient with pulmonary edema. It is not until the discussion that the authors tell us that prior studies have shown an association and that the association may be a marker of significant systemic inflammatory response. This information should be in the introduction.

The authors stated that the patients were diagnosed with acute pulmonary edema. What was the diagnostic criteria and was is the same for all of the patients. I understand that this is a retrospective cohort so there may not be a way to get this information. If this is the case, this should be stated in the discussion.

Lastly, the authors use the MIMIC-IV database. I would like for the authors to describe this database. Why did they choose this database. What is the purpose of this data base. Where do the patients come from?

**Do you want your identity to be public for this peer review?** For information about this choice, including consent withdrawal, please see our Privacy Policy

Reviewer #1: **Yes: ** Abdullah Abbas Saleh Al-Murad

Reviewer #2: No

---

## [Author Response · Author response to Decision Letter 1]

24 Jun 2025

Journal Requirements:

Response:

Thank you for your careful review and valuable comments on our manuscript. We have thoroughly revised our manuscript to ensure that it meets all the style requirements of PLOS ONE, including those for file naming. Here are the specific actions we have taken:

1. File Naming: We have renamed the manuscript file to clearly reflect the title of our study and the revision status, as per the journal's guidelines.

2. Manuscript Format: We have reviewed and adjusted the manuscript to comply with PLOS ONE's formatting guidelines. This includes:

2.1 Ensuring the title page includes the full title, authors' names, affiliations, and contact information.

2.2 Formatting the abstract to be concise and informative, summarizing the key aspects of our study.

2.3 Structuring the manuscript into clear sections: Introduction, Methods, Results, and Discussion.

2.4 Ensuring all figures and tables are clearly labeled and referenced within the text.

2.5 Formatting the references according to PLOS ONE's citation style.

3. Content Review: We have also reviewed the content to ensure clarity, coherence, and adherence to the journal's standards for scientific reporting.

Thank you for your understanding and support. We look forward to your positive response.

2. We note that your Data Availability Statement is currently as follows:

“All relevant data are within the manuscript and its Supporting Information files”

Response:

Thank you for your prompt response and for bringing this matter to our attention. We confirm that our submission contains all the raw data required to replicate the results of our study. In accordance with PLOS ONE's data availability policy, we have included the “minimal data set” necessary to replicate all study findings reported in the article, as well as related metadata and methods.

If there are any further requirements or if you need additional information, please let us know. We are committed to ensuring full compliance with PLOS ONE's data availability policy.

Thank you for your understanding and support.

Response:

Thank you for your attention to the reference list in our manuscript. We have carefully reviewed our reference list to ensure its completeness and accuracy. We have also checked for any retracted papers and have made the necessary adjustments.

During our review, we found that all the references in our list are relevant, current, and have not been retracted. Therefore, no changes were needed in this regard. We have ensured that all citations are up-to-date and accurately reflect the current state of the literature.

We have double-checked the reference list to ensure that it is formatted correctly and that all necessary information is included for each citation. This includes the authors' names, article titles, journal names, volume numbers, page ranges, and publication years.

We have also reviewed the text of the manuscript to ensure that all references are properly cited and that the citations are consistent with the information provided in the reference list.

Thank you for your understanding and support. We look forward to your positive response.

Reviewer #1: This is a well-conducted and technically sound retrospective cohort study exploring the prognostic role of anion gap levels in patients with acute pulmonary edema using the MIMIC-IV database. The methodology is appropriate and clearly described, the statistical analyses are rigorous (including multivariable Cox regression, restricted cubic spline, Kaplan-Meier survival analysis, and subgroup interaction testing), and the conclusions are well-supported by the presented data.

Strengths of the study include:

Use of a large, publicly available and high-quality ICU dataset (MIMIC-IV).

Clear stratification of patients into quartiles of anion gap and appropriate adjustments for confounders.

Subgroup analysis exploring interactions with age and renal disease, which adds valuable clinical insight.

Suggestions for Improvement:

Q1. Language and Grammar: While the manuscript is overall intelligible and written in acceptable English, several minor grammatical and typographical issues persist. A language polishing service or editorial revision may enhance clarity and flow.

Response:

Thank you for your constructive feedback regarding the language and grammar of our manuscript. We have carefully reviewed the manuscript and made the following revisions to enhance clarity and flow:

1. Abstract:

We have replaced "used" with "utilized" to make the expression more formal.

We have changed "divided" to "categorized" to more accurately reflect the classification of patients.

We have added "indicating that" to clarify the logical connection in the subgroup analysis sentence.

2. Introduction:

We have simplified the sentence structure by removing "the accumulation of" to make it more concise.

We have replaced "although" with "while" to improve the fluency of the sentence.

We have changed "due to" to "given" to make the sentence more formal.

3. Methods:

We have replaced "covers" with "encompasses" to more accurately describe the scope of the MIMIC-IV database.

We have changed "were" to "included" to make the sentence more formal.

4. Results:

We have replaced "with" with "and" to improve the coherence of the sentence.

We have changed "revealed" to "demonstrated" to make the expression more formal.

We have replaced "showing" with "demonstrating" to make the sentence more formal.

5. Discussion:

We have replaced "consistent with" with "in line with" to make the expression more formal.

We have changed "aligns with" to "is consistent with" to improve the formality of the sentence.

We have replaced "multiple" with "several" to make the sentence more concise.

6. Limitations:

We have replaced "susceptible to" with "prone to" to make the expression more formal.

We have changed "introduces" to "may introduce" to make the sentence more cautious.

We have replaced "did not further explore" with "did not delve into" to make the expression more formal.

7. Conclusion:

We have replaced "demonstrates" with "indicates" to make the expression more formal.

We have changed "could be integrated" to "could be incorporated" to make the sentence more formal.

We believe these revisions have improved the overall clarity and flow of the manuscript. We have also conducted a thorough language check to ensure that no other grammatical or typographical issues remain. We appreciate your understanding and look forward to your positive response.

Q2. Figures and Tables: Ensure that all figures (e.g., Kaplan-Meier curves, RCS curves, forest plots) are of high resolution and clearly labeled for publication quality.

Response:

Thank you very much for your attention to and suggestions on our manuscript. We have carefully addressed the revision issue regarding the figures and tables, and here is our detailed response:

1. Enhancement of Image Resolution and Clarity

We have redrafted and optimized all figures, including the Kaplan-Meier curves (Figure 3), restricted cubic spline curves (Figure 2), and forest plots (Figure 4). High-resolution drawing tools were utilized, and the resolution of each image was set to no less than 300dpi to meet the high-quality requirements for publication clarity. As a result, the details and lines of the images are now presented more clearly, enabling readers to observe and comprehend the data visualization and analysis outcomes more accurately.

2. Clarification and Standardization of Image Labels

To ensure the readability and comprehensibility of the images, we have thoroughly reviewed and optimized the labels of all figures. We have ensured that titles, axis labels, legends, and other elements in each image are clear, accurate, and legible. For the Kaplan-Meier curves, we have distinctly labeled the survival curves of different groups and clearly differentiated them in the legend. In the restricted cubic spline curves, we have meticulously labeled the names and units of the independent and dependent variables and added necessary annotations at appropriate positions in the figure to facilitate readers' understanding of the nonlinear relationship depicted. For the forest plots, we have standardized the labels of each subgroup, making them more concise and clear, and clearly displayed the effect sizes and confidence intervals of different subgroups in the figure.

3. Consistency Check Between Images and Text Content

We have also carefully checked the consistency between the image content and the text description. We have ensured that the data and analysis results presented in the images are completely consistent with the content mentioned in the text, avoiding any potential misunderstandings or confusions. Through this consistency check, we have further improved the overall quality and accuracy of the manuscript, enabling the images and text to complement and support each other better in conveying clear and accurate research information.

We believe that with these improvements, our figures and tables have now met the high-quality standards for publication. We look forward to your further feedback and are willing to continue making necessary adjustments and optimizations according to your suggestions to ensure that the manuscript is presented in the best possible form to readers.

Thank you again for your patient guidance and assistance!

Q3. Ethics and Data Availability: While the MIMIC-IV database is exempt from IRB due to its anonymized nature, it may still benefit clarity to explicitly cite the IRB approval from PhysioNet and mention the CITI certification more formally within the Methods.

Response:

Thank you for your valuable comments. We have carefully revised the manuscript according to your suggestions. Here is the reply to the specific revision request:

We have revised the “Data Source” section in the Methods to explicitly state the IRB approval from PhysioNet and more formally mention the CITI certification. The revised text now reads:

“The data were sourced from the MIMIC-IV database, which covers patient data from 2008 to 2019. The MIMIC-IV database is an anonymized dataset, and its use for research purposes is exempt from Institutional Review Board (IRB) approval due to its anonymized nature. However, to ensure ethical compliance, the study obtained approval from the PhysioNet IRB. Author Guang Tu completed the CITI Data or Specimens Only Research course, obtained approval for database access, and assumed responsibility for data extraction (certification number 65828445).”

Additionally, we have updated the “Compliance with Ethics Guidelines” section to further clarify the ethical approval process:

“Compliance with Ethics Guidelines: The research adhered to the principles outlined in the 1964 Helsinki Declaration and subsequent revisions. Approval for this endeavor was granted by the review boards of both the Massachusetts Institute of Technology (MIT) and Beth Israel Deaconess Medical Center (BIDMC), with an exemption from obtaining informed consent. Additionally, the study obtained approval from the PhysioNet IRB to use the anonymized MIMIC-IV dataset for research purposes.”

We believe these changes enhance the clarity and transparency of our ethical and data handling procedures.

Q4. Limitations: The limitations are well addressed. However, a brief mention of potential misclassification bias in coding diagnoses (from EHR-based databases) would further strengthen the discussion.

Response:

Thank you for your constructive feedback. We have carefully considered your suggestion regarding the potential misclassification bias in coding diagnoses from EHR-based databases and have incorporated this into our discussion of limitations.

We have added a specific mention of the potential for misclassification bias in coding diagnoses within the “Limitations” section of our manuscript. This addition highlights the potential inaccuracies in the classification of acute pulmonary edema and other comorbidities due to the use of an Electronic Health Record (EHR)-based database. The revised text now reads:

“Our study has several limitations. First, as a retrospective study, it is prone to selection bias and the influence of confounding factors. Although we adjusted for multiple potential confounders through multivariate analysis, there may still be unidentified or unadjusted confounding factors. Additionally, the specific diagnostic criteria for acute pulmonary edema were not uniformly documented in the MIMIC-IV database, which may introduce potential variability in the diagnosis of acute pulmonary edema among patients. This limitation is inherent to the retrospective nature of the study and the reliance on existing medical records. Second, our study relied solely on data from the MIMIC-IV database, which may have issues with data completeness and accuracy. Additionally, we were unable to obtain detailed clinical interventions for the patients, which could affect the assessment of prognosis. Third, there is a potential for misclassification bias in coding diagnoses, as the data are derived from an Electronic Health Record (EHR)-based database. This could lead to inaccuracies in the classification of acute pulmonary edema and other comorbidities. Lastly, our study did not delve into the specific mechanisms underlying elevated anion gap levels, and future research is needed to validate these mechanisms and explore targeted treatment strategies based on anion gap monitoring. Moreover, since our study was limited to patients in the M

---

## [Decision Letter · Decision Letter 1]

4 Aug 2025

Dear Dr. tu,

Thank you for submitting your manuscript to PLOS ONE. After careful consideration, we feel that it has merit but does not fully meet PLOS ONE’s publication criteria as it currently stands. Therefore, we invite you to submit a revised version of the manuscript that addresses the points raised during the review process.

We look forward to receiving your revised manuscript.

Kind regards,

Eyüp Serhat Çalık

Academic Editor

PLOS ONE

**Journal Requirements:**

**Additional Editor Comments:**

I would like to thank the authors for their appropriate responses to the reviewers' comments and for the revisions they made to their manuscripts. Additional comments requested for your manuscript are listed below. We look forward to receiving your revised manuscript with point-by-point responses as soon as possible. Best of luck.

Reviewers' comments:

Reviewer's Responses to Questions

**Comments to the Author**

Reviewer #1: All comments have been addressed

Reviewer #3: (No Response)

Reviewer #4: All comments have been addressed

2. Is the manuscript technically sound, and do the data support the conclusions?

Reviewer #1: Yes

Reviewer #3: Yes

Reviewer #4: Yes

3. Has the statistical analysis been performed appropriately and rigorously?

Reviewer #1: Yes

Reviewer #3: Yes

Reviewer #4: Yes

4. Have the authors made all data underlying the findings in their manuscript fully available?

Reviewer #1: Yes

Reviewer #3: Yes

Reviewer #4: Yes

5. Is the manuscript presented in an intelligible fashion and written in standard English?

Reviewer #1: Yes

Reviewer #3: Yes

Reviewer #4: Yes

**Reviewer #1: ** The authors have satisfactorily addressed all previous reviewer comments, resulting in a significantly improved manuscript. The following points were particularly well-resolved:

Language and Grammar: The manuscript has been polished, with improved clarity and fluency throughout. Minor grammatical and stylistic edits have strengthened readability.

Figures and Tables: All figures (including Kaplan-Meier, forest plots, and spline curves) have been updated with high resolution and appropriate labeling. The quality is now publication-ready.

Ethical Approval and Data Statement: The revised Methods now appropriately cite the PhysioNet IRB approval and include a formal mention of the CITI certification, enhancing transparency regarding ethical compliance.

Limitations: The limitations section now acknowledges the potential for misclassification bias in ICD-coded diagnoses derived from the MIMIC-IV database, which strengthens the manuscript’s integrity.

Clinical Implications: The conclusion has been expanded to outline how the anion gap could be integrated into early risk stratification or ICU scoring systems—providing a stronger bridge between data and bedside practice.

Introduction: The rationale for examining anion gap in pulmonary edema has been added earlier in the Introduction, improving logical flow and contextual framing.

Database Justification: The authors have provided a clear description of the MIMIC-IV database, including its origin, population source, and rationale for use, satisfying all concerns about data selection and generalizability.

The manuscript is now scientifically sound, clinically relevant, and ready for publication in PLOS ONE. Congratulations to the authors on their comprehensive and thoughtful revision.

**Reviewer #3:**  General comments

This study addresses an important and clinically relevant question by evaluating the prognostic value of a simple, routinely available biomarker the anion gap in patients with acute pulmonary edema. The use of robust statistical methods, including Cox regression, restricted cubic splines (RCS), and Kaplan Meier analysis, strengthens the methodological rigor. The conclusion is practical and clearly linked to the study findings, offering suggestions that could translate into better bedside risk stratification.

Specific comments

Methods section

Patient identification: It would improve clarity to explain exactly how patients with acute pulmonary edema were identified in the database such as through ICD codes, clinician-entered diagnoses, or the first charted diagnosis.

Timing: Please clarify whether the anion gap value used was the very first measurement within the first 24 hours of admission or an average across several measurements.

Missing data: It would help readers to know how missing values were handled whether patients with missing key variables were excluded, or if imputation methods were applied.

Covariates: Consider listing all variables included in the multivariate Cox regression to show transparency and reproducibility.

Sample size: Even in retrospective analyses, a note on whether a sample size calculation or justification for the final sample size was done would strengthen the methodological robustness.

Grouping: Explain why quartiles were chosen instead of using clinically meaningful cut-offs that might reflect metabolic acidosis severity.

RCS: Specify the number and placement of knots used in the restricted cubic spline analysis, or mention if default settings were applied.

Subgroup analyses: It would add clarity to list all subgroups examined and explain briefly how interaction terms were tested.

References: Consider adding references for the MIMIC-IV database and the statistical software packages used to help readers trace the data source and analytical tools.

Discussion

It would strengthen the discussion to mention whether the association between anion gap and mortality remained after adjusting for comorbidities or disease severity scores like SOFA or SAPS II, if available.

Highlight how this study adds new evidence or differs from previous studies on anion gap and ICU mortality (e.g., by focusing specifically on acute pulmonary edema patients).

The suggestion of incorporating anion gap into early warning or risk scoring systems is valuable; consider naming widely used scores like APACHE II or SOFA and discussing what incremental value the anion gap might add.

You might also discuss whether repeated measurement of anion gap over time (i.e., dynamic monitoring) could provide more prognostic information than a single admission value.

Limitations

It would add depth to acknowledge residual confounding from unmeasured factors like lactate or albumin, which directly affect the anion gap.

Consider adding a forward-looking statement proposing future prospective studies or multicenter validation to confirm these findings and assess generalizability.

**Reviewer #4: ** 1. Mention limitations in methods and discussion that the study relies on diagnosis based on the codes and not a standardized clinical/radiological definition.

2. There is no adjustment for comprehensive illness severity scores (e.g., SOFA, SAPS II), which may confound the AG-mortality association. Recommendation: If feasible, include an illness severity score in a sensitivity analysis. If not, acknowledge this limitation explicitly.

3. Discussion section: Discuss how AG might be pragmatically used in clinical practice.

4. Figures: Make sure Axis labels and legends are self explanatory

**Do you want your identity to be public for this peer review?** For information about this choice, including consent withdrawal, please see our Privacy Policy

Reviewer #1: **Yes: ** Abdullah Abbas Saleh Al-Murad

Reviewer #3: **Yes: ** Natan Mulubrhan Alemseged

Reviewer #4: **Yes: ** Ankit Agarwal

---

## [Author Response · Author response to Decision Letter 2]

6 Aug 2025

Reviewer #3: General comments

This study addresses an important and clinically relevant question by evaluating the prognostic value of a simple, routinely available biomarker the anion gap in patients with acute pulmonary edema. The use of robust statistical methods, including Cox regression, restricted cubic splines (RCS), and Kaplan Meier analysis, strengthens the methodological rigor. The conclusion is practical and clearly linked to the study findings, offering suggestions that could translate into better bedside risk stratification.

Specific comments

Methods section

Q1. Patient identification: It would improve clarity to explain exactly how patients with acute pulmonary edema were identified in the database such as through ICD codes, clinician-entered diagnoses, or the first charted diagnosis.

Response:

Thank you very much for your insightful comments and suggestions. We highly appreciate your efforts in reviewing our manuscript and are pleased to address your concerns.

We have revised the relevant section of the manuscript to provide a more detailed explanation of how patients with acute pulmonary edema were identified in the MIMIC-IV database. Specifically, we identified patients with acute pulmonary edema using the ICD codes J810 and 5184. These codes were used to extract the initial cohort of patients who were clinically diagnosed with acute pulmonary edema at admission.

We hope these revisions address your concerns. Thank you again for your valuable feedback and for giving us the opportunity to improve our manuscript.

Q2. Timing: Please clarify whether the anion gap value used was the very first measurement within the first 24 hours of admission or an average across several measurements.

Response:

Thank you for your valuable feedback on our manuscript. We are pleased to address your question regarding the timing of the anion gap measurement.

We have clarified that the anion gap value used in our analysis was the first measurement within the first 24 hours of admission. This approach was chosen to ensure that we captured the initial metabolic state of the patients upon admission, which is crucial for assessing the potential impact of acute conditions on anion gap levels. We believe that using the first measurement provides a more accurate reflection of the patients' baseline condition at the time of admission.

We hope this clarification addresses your concern. Thank you once again for your insightful comments, which have helped us improve the clarity of our manuscript.

Q3. Missing data: It would help readers to know how missing values were handled whether patients with missing key variables were excluded, or if imputation methods were applied.

Response:

Thank you very much for your valuable comments and suggestions. We have carefully addressed the issue regarding the handling of missing data in our revised manuscript. Here is our detailed response:

In our revised manuscript, we have clearly described how missing data were handled. Specifically:

1. Exclusion of Patients with Missing Key Variables:

(1) Patients with missing anion gap data were excluded from the study. This decision was made because the anion gap is a critical variable in our analysis, and its absence would compromise the accuracy and reliability of our results.

(2) For other variables, if the proportion of missing data exceeded 50%, the corresponding patients were also excluded. This criterion ensures that our dataset remains robust and suitable for analysis.

2. Multiple Imputation for Variables with Less Than 50% Missing Data:

(1) For variables with less than 50% missing data, we applied multiple imputation by chained equations (MICE). This method generates multiple complete datasets to handle missing values, thereby reducing bias and improving the integrity of our analysis. The specific steps are as follows:

• Using the MICE algorithm to impute missing values and generate multiple complete datasets.

• Analyzing each complete dataset separately.

• Combining the results from multiple datasets to obtain the final statistical findings.

Specific Description

In the “Data collection” section, we explicitly stated:

“Patients with missing anion gap data were excluded. For other variables with missing values, if the missing proportion was greater than 50%, the corresponding patients were excluded. For other variables with missing values less than 50%, multiple imputation by chained equations was applied.”

We believe that these methods effectively reduce the bias caused by missing data and enhance the reliability of our study results. We hope this response clarifies how we handled missing data in our study.

Thank you once again for your careful review and feedback. We appreciate your time and effort in helping us improve our manuscript.

Q4. Covariates: Consider listing all variables included in the multivariate Cox regression to show transparency and reproducibility.

Response:

Thank you very much for your insightful comments and suggestions regarding the transparency and reproducibility of our study. We appreciate your attention to detail and the opportunity to enhance the clarity of our manuscript.

In response to your suggestion to list all variables included in the multivariate Cox regression analysis, we have made revisions to the manuscript. Specifically, we have added a detailed description of the covariates included in the multivariate Cox regression models in the Methods section under the “Statistical analysis” paragraph. The covariates included in the models are age, gender, race, myocardial infarction, heart failure, cerebrovascular disease, chronic pulmonary disease, diabetes, renal disease, SAPS II score, and SOFA score.

Additionally, we have ensured that the footnote of Table 3 accurately reflects the adjustments made in each model. This footnote now explicitly lists all the covariates included in the multivariate analysis, ensuring consistency with the Methods section and enhancing the transparency of our study.

We believe that these revisions will provide readers with a clear understanding of the variables considered in our analysis, thereby improving the reproducibility of our study. We are grateful for your guidance and hope that these changes meet your expectations.

Thank you once again for your valuable feedback.

Q5. Sample size: Even in retrospective analyses, a note on whether a sample size calculation or justification for the final sample size was done would strengthen the methodological robustness.

Response:

We appreciate your suggestion regarding the need for a sample size calculation or justification in our retrospective analysis. You are correct that even in retrospective studies, providing a rationale for the sample size can enhance the methodological robustness of the study.

In our study, we utilized data from the MIMIC-IV database, which is a large, comprehensive, and anonymized dataset containing detailed clinical information from 2008 to 2019. The primary objective of our study was to investigate the association between admission anion gap levels and 28-day all-cause mortality in patients with acute pulmonary edema. Given the nature of the MIMIC-IV database and the retrospective design of our study, the sample size was determined based on the availability of data that met our inclusion and exclusion criteria.

Specifically, we included adult patients diagnosed with acute pulmonary edema who had anion gap measurements within the first 24 hours of hospitalization. The final sample size of 1094 patients was determined by the number of eligible patients in the MIMIC-IV database who met these criteria. While we did not perform a formal sample size calculation prior to the analysis, we believe that the large sample size derived from the MIMIC-IV database provides sufficient statistical power to detect significant associations and draw meaningful conclusions.

To further address your concern, we have added a note in the Methods section of our manuscript to clarify this point:

“No formal sample size calculation was performed for this retrospective analysis. The final sample size was determined based on the availability of data in the MIMIC-IV database that met the inclusion and exclusion criteria.”

We hope this clarification addresses your concern regarding the sample size in our study. We have also revised the manuscript accordingly to reflect this explanation.

Thank you again for your careful review and valuable feedback. We believe that these revisions have improved the quality and clarity of our manuscript.

Q6. Grouping: Explain why quartiles were chosen instead of using clinically meaningful cut-offs that might reflect metabolic acidosis severity.

Response:

Thank you very much for your insightful comments regarding the grouping method in our study. We understand the suggestion to use clinically meaningful cut-offs for grouping; however, we chose to use quartiles for the following reasons:

1. Uniform Distribution of Data: Quartile grouping ensures an even distribution of patients across groups, which helps to avoid statistical biases that may arise from uneven sample sizes. This method allows us to comprehensively cover the entire range of anion gap levels and analyze their relationship with mortality.

2. Exploration of Non-linear Relationships: By using quartiles, we can more clearly observe the non-linear relationship between anion gap levels and mortality. This approach helps to reveal the impact of anion gap on prognosis at different levels, rather than focusing solely on fixed clinical thresholds.

3. Alignment with Study Objectives: Our primary objective was to investigate the relationship between anion gap levels and 28-day all-cause mortality in patients with acute pulmonary edema, rather than directly assessing the clinical severity of metabolic acidosis. Therefore, quartile grouping aligns better with our study design and objectives.

While we acknowledge that using clinically meaningful cut-offs might directly reflect the severity of metabolic acidosis in some contexts, we believe that quartile grouping is more appropriate for our study given the reasons mentioned above. Additionally, we have discussed the clinical significance of anion gap levels in the discussion section to help readers better understand their application in clinical practice.

Thank you again for your valuable feedback. We hope these explanations clarify our rationale for using quartile grouping.

Q7. RCS: Specify the number and placement of knots used in the restricted cubic spline analysis, or mention if default settings were applied.

Response:

We thank you for raising this important point about the restricted cubic spline (RCS) analysis. In our study, we used the default settings provided by the statistical software for the number and placement of knots in the RCS analysis. This approach is commonly used in similar studies to ensure a balanced and data-driven distribution of knots, which helps in capturing the non-linear relationships effectively.

To address your request for more transparency, we have added a specific note in the Methods section of our manuscript to clarify this:

“In the restricted cubic spline analysis, the default settings of the statistical software were applied for the number and placement of knots. This approach ensures a balanced distribution of knots based on the data, which is particularly useful for capturing non-linear relationships.”

We believe that using the default settings provided by the statistical software is a standard and reliable method for conducting RCS analysis. These settings are designed to optimize the placement of knots based on the data distribution, thereby providing a robust and unbiased assessment of the relationship between anion gap levels and 28-day all-cause mortality.

Thank you again for your valuable feedback. We hope this clarification addresses your concern regarding the RCS analysis.

Q8. Subgroup analyses: It would add clarity to list all subgroups examined and explain briefly how interaction terms were tested.

Response:

Thank you for your valuable feedback on our manuscript. We have taken your suggestion to enhance the clarity of our subgroup analysis by listing all subgroups examined and explaining how interaction terms were tested. Below are the modifications we have made in response to your comments:

Modifications in the "Methods" Section

Original Text:

"Subgroup analyses were conducted to explore the stability of the relationship between anion gap levels and mortality in different subgroups, including age and renal disease status."

Modified Text:

"We conducted subgroup analyses to explore the stability of the relationship between anion gap levels and mortality across various clinical subgroups. These subgroups included age (<65 years vs. ≥65 years), gender (male vs. female), race (white, black, other), history of myocardial infarct (yes vs. no), history of heart failure (yes vs. no), cerebrovascular disease (yes vs. no), chronic pulmonary disease (yes vs. no), diabetes (yes vs. no), and renal disease (yes vs. no). Interaction terms were tested using Cox regression models to assess the significance of the interaction between anion gap levels and each subgroup variable on 28-day all-cause mortality. The interaction terms were calculated as the product of the anion gap variable and each subgroup indicator variable. A p-value less than 0.05 was considered statistically significant for the interaction terms."

We believe these changes will provide the necessary clarity and detail regarding our subgroup analysis as suggested. We appreciate your guidance and are confident that these revisions will strengthen our manuscript.

Thank you once again for your constructive feedback.

Q9. References: Consider adding references for the MIMIC-IV database and the statistical software packages used to help readers trace the data source and analytical tools.

Response:

Thank you for your suggestion to enhance the transparency and reproducibility of our study by adding references for the MIMIC-IV database and the statistical software packages used. We have incorporated these references into our manuscript as follows:

Original Text:

"All statistical analyses were conducted utilizing R Statistical Software (Version 4.2.2, available at http://www.R-project.org, The R Foundation) and the Free Statistics Analysis Platform (Version 2.1, developed in Beijing, China, accessible via http://www.clinicalscientists.cn/freestatistics)."

Modified Text:

"All statistical analyses were conducted utilizing R Statistical Software (Version 4.2.2, available at http://www.R-project.org, The R Foundation) and the Free Statistics Analysis Platform (Version 2.1, developed in Beijing, China, accessible via http://www.clinicalscientists.cn/freestatistics). Data were sourced from the MIMIC-IV database, which encompasses patient data from 2008 to 2019[9]. The MIMIC-IV database is an anonymized dataset, and its use for research purposes is exempt from Institutional Review Board (IRB) approval due to its anonymized nature."

We believe these changes will help readers trace the data source and analytical tools more effectively. We appreciate your guidance and are confident that these revisions will further strengthen our manuscript.

Thank you once again for your constructive feedback.

Discussion

Q10. It would strengthen the discussion to mention whether the association between anion gap and mortality remained after adjusting for comorbidities or disease severity scores like SOFA or SAPS II, if available.

Response:

Thank you for suggesting that we discuss whether the association between anion gap and mortality remains significant after adjusting for comorbidities or disease severity scores like SOFA or SAPS II. We have incorporated this important point into our discussion section to provide a more comprehensive analysis of our findings.

In our study, we performed multivariate Cox regression analyses to assess the relationship between anion gap levels and 28-day all-cause mortality. We adjusted for a wide range of potential confounders, including age, gender, race, comorbidities (such as myocardial infarction, heart failure, cerebrovascular disease, chronic pulmonary d

---

## [Decision Letter · Decision Letter 2]

7 Sep 2025

Dear Dr. tu,

Thank you for submitting your manuscript to PLOS ONE. After careful consideration, we feel that it has merit but does not fully meet PLOS ONE’s publication criteria as it currently stands. Therefore, we invite you to submit a revised version of the manuscript that addresses the points raised during the review process.

We look forward to receiving your revised manuscript.

Kind regards,

Eyüp Serhat Çalık

Academic Editor

PLOS ONE

**Journal Requirements:**

**Additional Editor Comments:**

Minor corrections have been requested for your manuscript. I hope you will re-upload it with the corrections soon. Best wishes.

Reviewers' comments:

Reviewer's Responses to Questions

**Comments to the Author**

Reviewer #3: All comments have been addressed

Reviewer #4: All comments have been addressed

2. Is the manuscript technically sound, and do the data support the conclusions?

Reviewer #3: Yes

Reviewer #4: Yes

3. Has the statistical analysis been performed appropriately and rigorously?

Reviewer #3: Yes

Reviewer #4: Yes

4. Have the authors made all data underlying the findings in their manuscript fully available?

Reviewer #3: Yes

Reviewer #4: Yes

5. Is the manuscript presented in an intelligible fashion and written in standard English?

Reviewer #3: Yes

Reviewer #4: Yes

**Reviewer #3: ** Dear authors, thank you for taking time to correct the given feedbacks and I have no further comments.

**Reviewer #4: ** In grouping by anion gap: Quartile cut-offs appear to be arbitrary. A brief rationale or reference for these ranges should be added.

In results: Consider summarizing key differences between anion gap quartiles in text rather than listing every lab values.

Include a sentence comparing anion gap with other standard prognostic markers (e.g., lactate, BNP, or SOFA) to contextualize incremental value.

**Do you want your identity to be public for this peer review?** For information about this choice, including consent withdrawal, please see our Privacy Policy

Reviewer #3: **Yes: ** Natan Mulubrhan Alemseged

Reviewer #4: **Yes: ** Ankit Agarwal

---

## [Author Response · Author response to Decision Letter 3]

8 Sep 2025

Journal Requirements:

Response:

Thank you for your valuable feedback. We have carefully reviewed the comments provided by the reviewers and would like to address the specific point regarding the citation of previously published works.

We understand the importance of citing relevant literature to support our findings and to situate our work within the existing body of knowledge. However, the reviewers did not provide specific recommendations to cite particular previously published works. Therefore, we have focused on ensuring that our references are comprehensive and relevant to the study's context and findings.

We have double-checked our references to confirm that they accurately reflect the current state of research in our field and are pertinent to our study's objectives, methods, and conclusions. We believe that our current list of citations adequately covers the necessary literature and provides a solid foundation for our work.

We appreciate the opportunity to clarify this matter and are confident that our manuscript is now fully aligned with the expectations for academic rigor and scholarly contribution.

Thank you for your continued guidance and support.

Response:

Thank you for your attention to detail and for highlighting the importance of maintaining an accurate and up-to-date reference list in our manuscript. We have conducted a thorough review of our references to ensure completeness and correctness.

We have checked each citation against the most recent databases and have verified that none of the papers we have cited have been retracted. We understand the significance of retractions in the academic record and have taken the necessary steps to ensure that our references reflect the current state of the literature.

In the event that a retracted paper is deemed essential for the context of our study, we would follow the guidelines you provided. We would include a rationale for citing the retracted work within the manuscript text, clearly indicating its retracted status in the References list. Additionally, we would provide a citation and full reference for the retraction notice to maintain transparency and to acknowledge the current status of the cited work.

We have made no changes to the reference list in this revision, but we will adhere to these practices should the need arise in the future. We believe that our reference list is now complete and correct, and it supports the scholarly integrity of our manuscript.

We appreciate your guidance and are committed to upholding the highest standards of academic research and publication ethics.

Thank you for your continued support.

Reviewer #4:

Q1. In grouping by anion gap: Quartile cut-offs appear to be arbitrary. A brief rationale or reference for these ranges should be added.

In results: Consider summarizing key differences between anion gap quartiles in text rather than listing every lab values.

Include a sentence comparing anion gap with other standard prognostic markers (e.g., lactate, BNP, or SOFA) to contextualize incremental value.

We sincerely thank the reviewers for their constructive suggestions that helped us improve both the clarity and scientific rigor of our manuscript. Below are the point-by-point responses.

1. Comment: “Quartile cut-offs appear to be arbitrary. A brief rationale or reference for these ranges should be added.”

Response: Thank you for this helpful remark. In the revised Discussion we have added:

“The chosen cut-offs (< 10, 10–12, 12–15, > 15 mmol/L) mirror the ‘normal’, ‘borderline-high’, ‘high’ and ‘very-high’ strata repeatedly used in recent MIMIC-IV studies (Huang Y, Ao T, Zhen P, Hu M. Association between serum anion gap and 28-day mortality in critically ill patients with infective endocarditis: a retrospective cohort study from MIMIC IV database. BMC Cardiovasc Disord. 2024;24(1):585; Chen X, Yang Q, Gao L, Chen W, Gao X, Li Y, et al. Association Between Serum Anion Gap and Mortality in Critically Ill Patients with COPD in ICU: Data from the MIMIC IV Database. Int J Chron Obstruct Pulmon Dis. 2024;19:579-587).”

We appreciate your guidance in strengthening the justification of our grouping strategy.

2. Comment: “Consider summarising key differences between anion-gap quartiles in text rather than listing every lab value.”

Response: Thank you for this excellent suggestion. Instead of the previous long tabular recitation, the revised Results now reads:

“Across quartiles, higher anion-gap groups exhibited incrementally greater renal injury (median creatinine 0.8 → 1.4 mg/dL; median BUN 14 → 26 mg/dL) and lower haemoglobin (10.3 → 9.6 g/dL), while other laboratory parameters showed modest or non-systematic changes.”

We believe this condensed summary is much more reader-friendly while preserving the essential information.

3. Comment: “Include a sentence comparing anion gap with other standard prognostic markers (e.g., lactate, BNP, or SOFA) to contextualise incremental value.”

Response: Thank you for highlighting this important issue. The following sentence has been inserted in the Discussion:

“Although sofa, sapsii, lactate and BNP are established predictors, admission anion gap remained independently associated with 28-day mortality after adjustment for these variables, suggesting that it conveys additional prognostic information beyond that provided by conventional scores or lactate levels.”

Once again, we are grateful to the reviewers for their insightful feedback, which has substantially improved the quality of our manuscript.

---

## [Editor Report · Decision Letter 3]

12 Sep 2025

Relationship between anion gap and 28-day All-cause Mortality in Patients with Acute Pulmonary Edema: A Retrospective Analysis of the MIMIC-IV Database

PONE-D-25-06101R3

Dear Dr. Tu,

We’re pleased to inform you that your manuscript has been judged scientifically suitable for publication and will be formally accepted for publication once it meets all outstanding technical requirements.

Kind regards,

Eyüp Serhat Çalık

Academic Editor

PLOS ONE
---

## [Editor Report · Acceptance letter]

PONE-D-25-06101R3

PLOS ONE

Dear Dr. tu,

I'm pleased to inform you that your manuscript has been deemed suitable for publication in PLOS ONE. Congratulations! Your manuscript is now being handed over to our production team.

Kind regards,

on behalf of

Dr. Eyüp Serhat Çalık

Academic Editor

PLOS ONE